# Sedimentary biomarkers of human presence and taro cultivation reveal early horticulture in Remote Oceania
Giorgia Camperio [1,2] ✉, S. Nemiah Ladd [3], Matiu Prebble [4,5], Ronald Lloren[1,2], Elena Argiriadis [6,7], Daniel B. Nelson [8], Christiane Krentscher[1] & Nathalie Dubois [1,2]

Remote Oceania was among the last places settled by humans. However, the timing of initial human settlements and the early introduction of horticulture remain debated. We retrieved a sediment core close to Teouma, the oldest cemetery in Remote Oceania that reveals evidence of initial settlement, horticulture practice, and concurrent climatic conditions on the island of Efate, Vanuatu. Sedimentary biomarkers indicating human presence (coprostanol and epicoprostanol), and taro cultivation (palmitone), increase simultaneously, attesting to the early introduction of horticulture by first settlers. The precipitation signal preserved in leaf waxes shows that the initial settlement occurred during a period of increasing wetness—climatic conditions favourable for the establishment of horticulture. The timing of these events is constrained by a high-resolution radiocarbon chronology that places the first unequivocal trace of human activity and horticulture at 2800 years ago. These findings advance our understanding of human history in the Pacific.

Remote Oceania, a region characterised by large inter-island distances, extends from the eastern Solomon Islands to the Polynesian triangle of the Hawaiian Islands, Rapa Nui and Aotearoa/New Zealand[1] (Fig. 1). These islands were on the oceanic pathways of multiple groups of horticulturalists seafarers throughout the last 3500 years. Early seafarers originated from the islands of Southeast Asia and Papua, while later migrations extended east to Polynesia[2–7]. The Lapita Cultural Complex, characterised by dentate-stamp ceramic ware, is associated with the earliest group to reach Remote Oceania[8–10] whose descendants founded the cultures of most of the Pacific Islands (Fig. 1). It has been suggested that during particular periods, climate windows may have favoured certain human migration routes[11–13], with adverse climatic conditions motivating land abandonment[14] or migrations to new islands[15]. In addition, past precipitation changes could have played a role in settlement patterns by influencing cultivation practices[16,17], driving cultural adaptations[18,19] and leading to differing migration routes at different times[15]. However, the resolution and chronologies of existing climatic records in Remote Oceania covering the period of Lapita settlement do not allow the actual influence of climate on human settlement patterns to be determined.

The archipelago of Vanuatu contains extensive archaeological remains of the Lapita Cultural Complex and is considered a key location in the initial settlement of Remote Oceania[20].

The islands of Vanuatu, and archipelagos to the East, were unoccupied prior to Lapita's arrival. Exact dating of early sites in Vanuatu associated with the Lapita is often complicated by intense perturbations due to post-depositional disturbance, inbuilt age, and reservoir offsets[21,22]. Although U/Th dating of coral artefacts has been used for precise dating of early settlements in Tonga[23] there are no such dates available for Vanuatu, which lies in the Western route of the Lapita migration. In addition, despite major improvements in radiocarbon techniques, these dates can still have large uncertainties, presenting challenges to developing a robust chronology of human settlement in the region[22,24]. Similarly, while there is archaeological evidence of agricultural practices in Vanuatu, such as terraced gardens[25] and introduced crops[26–28], identifying the precise timing and extent of early horticultural practices has been hindered by uncertainties in stratigraphy, dating, and biological proxies[29].

Given its archaeological importance for reconstructing the initial human settlement of Remote Oceania and its position at the south-western edge of the South Pacific Convergence Zone (SPCZ), Vanuatu is ideally

[1]Department of Surface Waters Research & Management, Eawag, Dübendorf, Switzerland. [2]Department of Earth Sciences, ETH Zürich, Zürich, Switzerland. [3]Department of Environmental Sciences, University of Basel, Basel, Switzerland. [4]School of Earth and Environment, College of Science, University of Canterbury, Christchurch, New Zealand. [5]Archaeology and Natural History, Culture History and Languages, The Australian National University, Canberra, Australia. [6]Institute of Polar Sciences, Venice, Italy. [7]Department of Environmental Sciences, Informatics and Statistics, Ca' Foscari University, Venice, Italy. [8]Department of Environmental Sciences—Botany, University of Basel, Basel, Switzerland. ✉e-mail: giorgia.camperio@eawag.ch

**Fig. 1 | Map of Emaotfer swamp on the island of Efate, Vanuatu in relation to the Lapita archaeological sites across remote Oceania. A** Map of the Lapita archaeological sites (red dots) on tropical Pacific islands[6]. Curved dashed line delimits Near Oceania and Remote Oceania[96]. Shading corresponds to monthly precipitation[97] (source of data https://disc.gsfc.nasa.gov/datasets/GPM_3IMERGM_06/summary). **B** Geological map of Efate (Vanuatu)[98]. Dots correspond to the archaeological sites discussed in the text, blue shading indicates the location of Emaotfer Swamp. **C** Map of the Teouma area with the Teouma archaeological site (red dot) and the Teouma river to the west of Emaotfer swamp, coring location shown in the swamp (map from https://en-gb.topographic-map.com/).

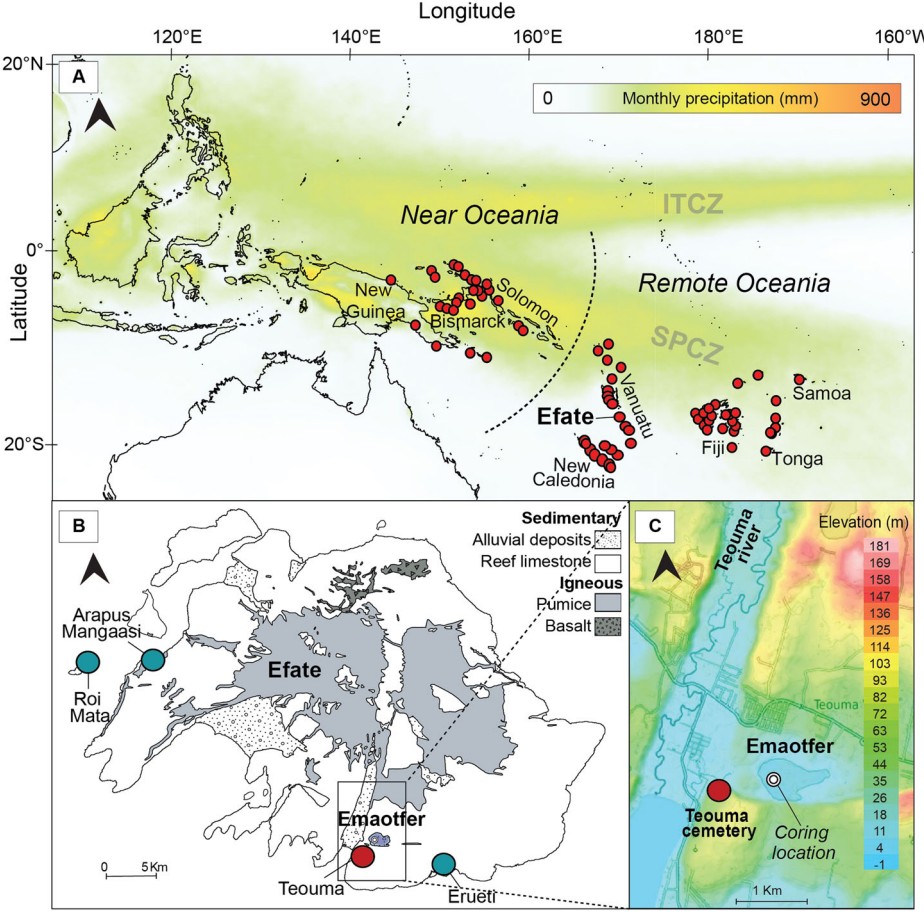

located to improve our understanding of the role of past climatic drivers on ancient settlements and migrations. Changes in precipitation are the main local climatic variable and are influenced by the Western Pacific Warm Pool (WPWP) and the SPCZ[30], both interconnected to the Intertropical Convergence Zone (ITCZ)[31]. Given the strong precipitation gradients that characterise the SPCZ, any shifts in its location and intensity can have far-reaching implications for island communities[32]. Interannual climate variability is influenced by the El Niño-Southern Oscillation (ENSO), which alters both the WPWP and SPCZ, causing droughts during El Niño events and wetter conditions during La Niña events[33,34]. ENSO-driven climate patterns can have considerable impacts on the availability of water resources and agricultural productivity[35] and may have affected early settlements.

Here, we track human arrival and hydroclimatic changes using lipid biomarkers from Emaotfer swamp sediments from the island of Efate, Vanuatu (Fig. 1). The swamp was the subject of previous palaeoecological investigations[36,37] and is located 1 km east of the Teouma archaeological site, the earliest dated Lapita cemetery in Remote Oceania. The Teouma site provides evidence for much of what we know about ancient Lapita food production, material culture, funerary practices and human genetic diversity[38,39]. To build on these findings, we use faecal biomarkers, namely coprostanol (5β-cholestan-3β-ol) and its epimer, epicoprostanol, as indicators of human presence. These molecules, which are produced by gut microbes as a metabolic product of cholesterol, are most abundant in human faeces[40] and have been used in archaeological contexts to reconstruct demographic changes[41,42]. Faecal biomarkers were coupled with palmitone (hentriacontan-16-one) which is a unique biomarker for taro (*Colocasia esculenta*), the main staple crop of the region introduced by early settlers[43,44]. Standard taro cultivation practices tend to prevent flowering and can limit pollen production, hindering identification of taro cultivation in sediments using conventional palynological analyses[45,46]. We circumvent these issues using palmitone which is not linked to pollen production[47]. Finally, to

understand the role of climate in the initial settlement of Remote Oceania and the establishment of horticulture, we use the compound-specific hydrogen isotopic composition of leaf wax long chain *n*-alkanoic acids from higher plants. These can reflect changes in the hydrogen isotopic composition of precipitation[48,49] if other effects such as post-rainfall evaporation/evapotranspiration and changes in plant communities are minimal (Supplementary text S1). At low-elevation, coastal sites such as the Emaotfer Swamp, hydrogen isotope ratios of precipitation are strongly linked to the amount of precipitation[50–52], allowing past hydroclimate to be reconstructed from leaf wax hydrogen isotope ratios.

This study employs a multiproxy approach by combining climatic proxies and biomarkers indicating human presence and taro cultivation, within a sedimentary record spanning the last 5000 years. Here, we establish a precise chronology of human presence at the Teouma site and contextualise the role of climate on early settlements and horticulture in Remote Oceania.

## Results
### Landscape dynamics

The archipelago of Vanuatu lies on the Pacific/Australian margin and is the result of emerged volcanic complexes and their interaction with Quaternary sea level changes and tectonic events, resulting in uplifted limestone massifs on most of the islands[53]. At ~6 m asl., the Teouma archaeological site is located on an uplifted limestone reef which emerged ~4000 years ago[54], following coastal uplift but also coincident with the end of the mid-Holocene sea-level highstand in the region[55]. Beach ridges that sit ~4 m asl are present 100–200 m inland of the current Teouma Bay coastline. These most likely represent the final phase of mid-Holocene sea level recession, and indicate that coastal uplift has not exceeded more than a few metres over the last 4000 years. U/Th dates on uplifted terraces across southern Efate show that uplift rates have not exceeded 1 mm/yr in the Holocene[53].

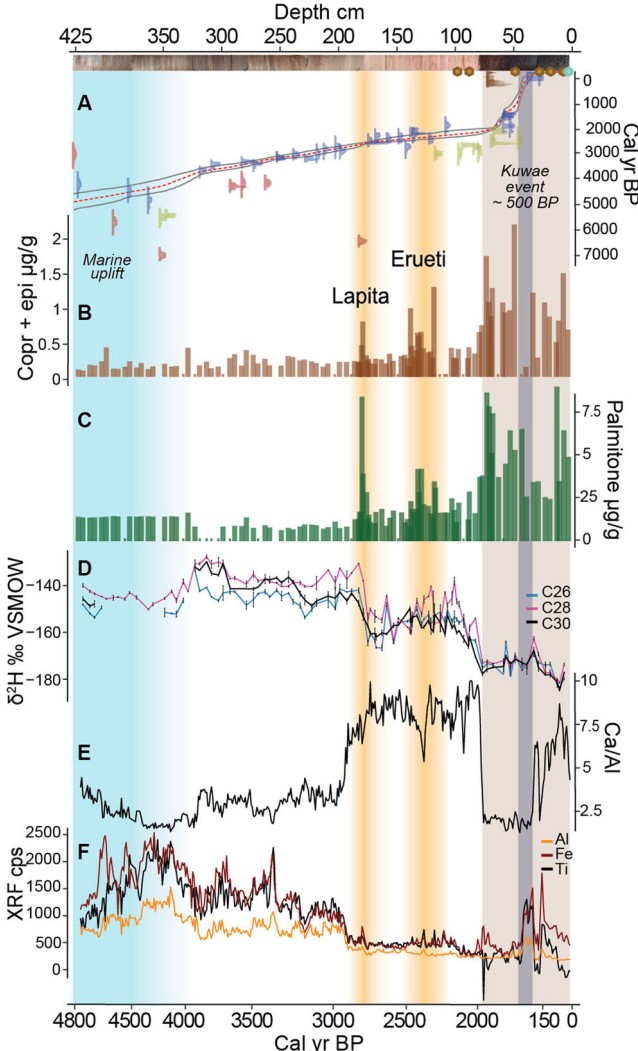

**Fig. 2 | Downcore fossil biomarker record of Efate tracking human presence, horticulture and paleoclimatic shifts along the paleoenvironmental record from XRF elemental counts. A** Age-depth model using the r package rbacon version 3.0.0[94]. Green hexagon for coring year (2017), brown hexagons for postbomb roots, brown age calibration pre-bomb roots. Yellow age calibration for samples that incorporated old carbon ($\delta^{13}C$ above—15‰). Blue plots represent the age distribution of macrofossils used in the age model, red for those that were excluded from the model. Grey lines indicate respectively the minimum and maximum range of calibrated ages. Red line indicates the median calibrated ages. **B** Coprostanol and epicoprostanol concentrations are indicated with brown bars, (**C**) palmitone with green bars both in µg/g of sediment. **D** Compound-specific *n*-alkanoic acids $\delta^2H$ in black (C30), in light pink (C28), and light blue (C26), in permil VSMOW. Changes in the elemental composition of the core measured with an XRF core scanner, with (**E**) the calcium to aluminium ratio (Ca/Al) in black and (**F**) terrigenous elements in black (titanium), red (iron), and orange (aluminium) in counts per seconds (cps). Light blue shading represents the estuarine period of the basin, yellow shadings represent periods of human occupation corresponding to the Lapita and Erueti phase in the archaeological record, brown shading indicates the peat part of the core, grey area indicates the possible location of the Kuwae volcanic tephra (1452 or 1453 CE)[99].

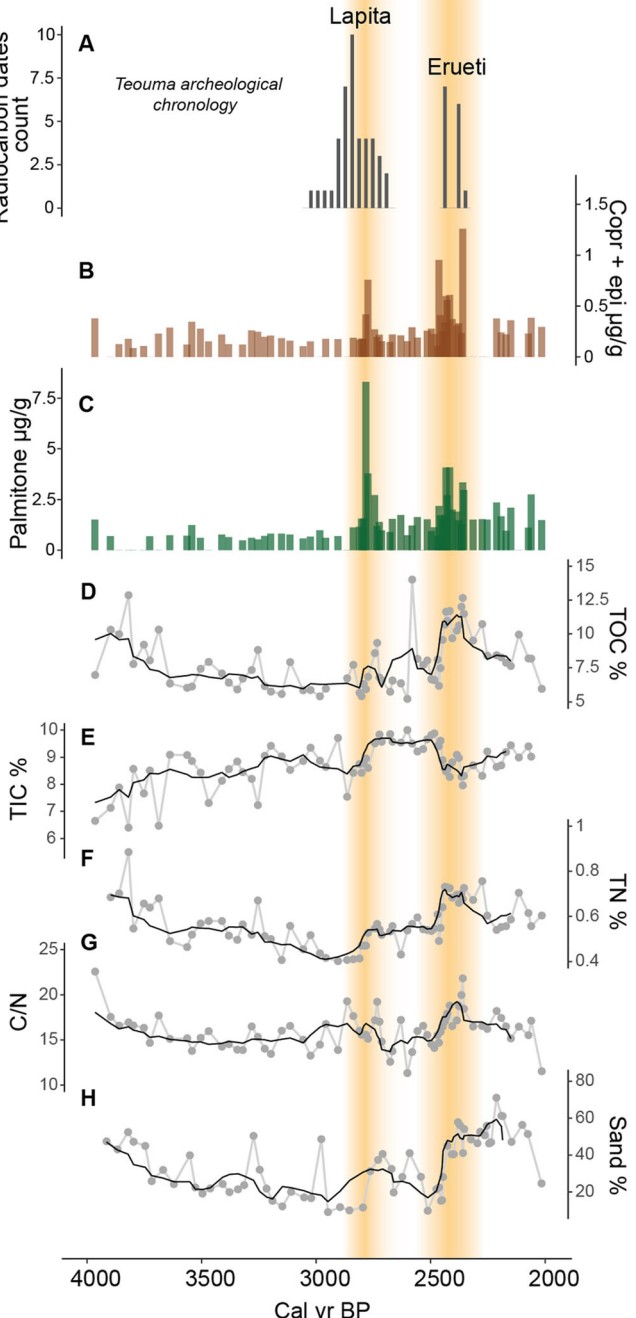

**Fig. 3 | Environmental changes during the Lapita and Erueti settlement. A** Histogram of median calibrated ages for the Lapita period combined with age range of the Erueti period unmodeled calibrated ages (68% prob)[22]. **B** Sum of coprostanol and epicoprostanol showing faecal markers concentrations as brown bars. **C** Palmitone concentrations as green bars. Black lines indicate five-point averages for (**D**) Total Organic Carbon (TOC), (**E**) Total Inorganic Carbon (TIC), (**F**) Total Nitrogen (TN), (**G**) C/N ratio, and (**H**) sand fraction. Yellow shades indicate periods of human occupation.

Emaotfer sediments record uplift and sea-level changes as a transition from a lagoon to a freshwater environment after ca. 4000 BP (Before Present; indicates calibrated years before 1950) (Fig. 2). The lagoon phase is identified by high $\delta^{15}N$ values (Fig. S1) and high terrigenous element counts (Fig. S2). The switch from a lagoon to lacustrine setting is supported by a decline in mangrove forest around the basin[51] and is associated with a transient peak in manganese (Mn) deposition (Fig. S3), suggesting a shift in redox conditions and a sudden deposition of Mn (Supplementary text S2).

These tectonic movements and sea level recession led to the emergence of new favourable land for first human settlement and the formation of a freshwater reservoir. The lacustrine phase lasted until the appearance of peat between 1792 and 2114 BP, characterised by lower C/N ratios (mean 16 ± 2, Fig. 3) and $\delta^{15}N$ values (mean 0.9 ± 0.3, Fig. S1). The lacustrine phase can be separated into two distinct intervals: An older one from 4000 BP to ca. 2900 BP, characterised by high values of terrigenous elements (Al, Fe, Ti), and a younger one from 2900 BP to 2000 BP, characterised by high values of the

Ca/Al ratio (Fig. 2). We attribute the early lacustrine phase to gradual infilling of the catchment, disconnecting any estuarine influence on the site either by blocking subterranean conduits below the limestone massif (common across Efate) or by the avulsion of the Teouma River to its current position further west of Emaotfer swamp (2811–2985 BP). The change in hydrology at 2900 BP is signalled by the sharp increase in the calcium to aluminium ratio (Ca/Al) linked with the final drop in terrigenous elements (Fig. 2).

Three major drivers, that may have a combined effect, can explain the hydrological shift in the swamp. First, tectonic movements causing an uplift could have rapidly altered the ground or surface water movement in the swamp. Emaotfer swamp today lies at a similar elevation as the Teouma Graben, between 2 and 5 m asl, but is separated from the Graben by higher grounds (Fig. 1C). The hydrological connections either via subterranean solutional conduits or through surface flow during flood events of the Teouma River may have been cut off after an uplift event. Second, the drier events preceding 2900 BP (Fig. 4) could have lowered the flow regime in the Emaotfer catchment. When conditions got wetter again, ground and surface water flows could have dramatically altered sedimentation patterns in the catchment. Finally, water management for cultivation could explain the sudden shift in hydrology. Numerous examples of sophisticated and large-scale horticultural production including irrigation and water management systems exist in the ethnographic and archaeological records of Efate and neighbouring islands[56–59]. Furthermore, shifting cultivation practices, still in use today, across the surrounding catchment would have resulted in the influx of calcium carbonate in soils eroded from the surrounding uplifted limestone massif as vegetation was cleared.

## Molecular tracers of human occupation

The proximity of Emaotfer swamp to the Teouma site facilitates a detailed comparison with the archaeological findings and specifically with the settlement chronology based on radiocarbon dating of shells and human remains[22,24]. The temporal precision provided by the sedimentary age model at Emaotfer, derived from 40 radiocarbon-dated samples (Fig. 2), is key to capturing the impact of Lapita settlement and provides a further constraint to the first period of human occupation of the Teouma site. High-resolution dating based on 26 samples provides a well-constrained age model between 2220 and 3860 cal BP, which includes the period of first human occupation (Fig. S4). The age reversal present at 181 cm (Fig. S4, table S1) could be indicative of land use during the early Lapita period as the result of human-induced erosion bringing older material in the swamp. Similar aged material from deeper parts of the core (and with similar $\delta^{13}C$ values, i.e. 351 cm) could indicate that old, preserved material could have been excavated during the early settlement. Age reversals in the marine sediment phase, root intrusions, and hard-water effects (identified by higher $\delta^{13}C$ values of eight macrofossils, table S1) at the peat transition result in higher age uncertainties in these two phases. Although age reversals in radiocarbon-dated macro-fossils were also encountered in a previous study from the Emaotfer swamp[36], the lower sampling resolution of this record, and more generally the age uncertainties of radiocarbon dating, limit the comparison between the two age models. While faecal sterols are present throughout the whole core at low and sometimes constant concentrations, indicating inputs from other non-human producers (Supplementary text S3), our study is uniquely situated to differentiate non-human from human-related coprostanol production because of the relatively recent and abrupt arrival of humans on remote Pacific islands. Sedimentary biomarkers associated with human presence (faecal biomarkers) and cultivation (palmitone) highlight three main periods of human occupation (Figs. 2 and 4) starting ca. 2800 BP until present.

The first period from 2624–2750 BP to 2739–2879 BP is short and coincident with the first Lapita settlement recorded at Teouma, which indeed would have only lasted a few generations[22]. The Bayesian chronological framework proposed by Petchey et al.[22] indicates 2870–2920 BP as the most likely start date of occupation of the site, but challenges related to cleaning, correction, and dietary considerations can limit the chronological

precision of the available archaeological materials. We used radiocarbon dating of short-lived material deposited in the Emaotfer sedimentary basin to further constrain the site chronology with a first unequivocal human trace appearing at 2739–2879 BP, at the youngest limit of the age (2870–2920 BP)

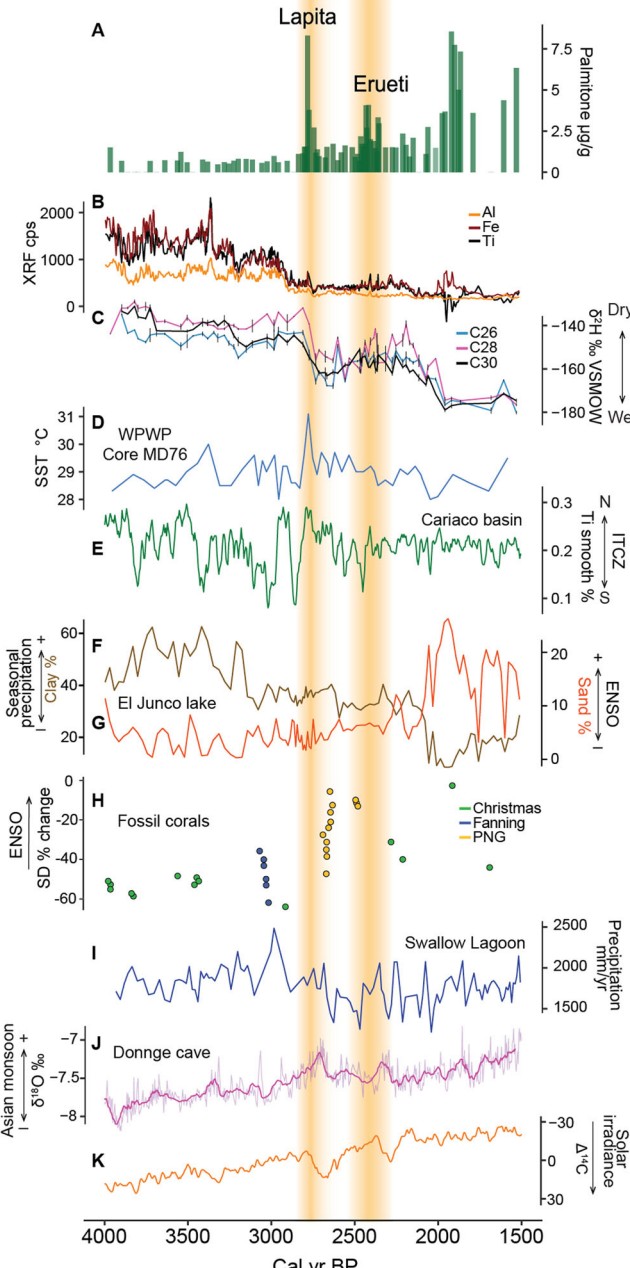

Fig. 4 | Climatic reconstructions for the period 4000–1500 BP across the Pacific. A Palmitone concentration (green bars). B XRF counts per second of terrigenous elements, black (titanium), red (iron), and orange (aluminium). C Compound-specific n-alkanoic acids $\delta^2H$ in black (C30), blue (C28), and light pink (C26), reported in per mil (‰) relative to VSMOW. Vertical black lines indicate standard deviations. D Western Pacific Warm Pool (WPWP) sea surface temperature reconstruction for core MD76 (light blue)[72]. E Titanium % from the Cariaco Basin in green[75]. Grain Size from El Junco Lake, Galapagos, with (F) clay (brown) and (G) sand (yellow)[77]. H Calculations of standard deviation of 2–7 year band of fossil coral $\delta^{18}O$ time series calculated as percentage difference from modern coral reference following Cobb et al.[80], Christmas Island[80,100] and its micro-atolls[101] in green, Fanning island in blue[80], and Papua New Guinea in gold[82,102]. I Mean annual precipitation from Swallow Lagoon[81]. J $\delta^{18}O$ from the Donnge Cave (pink)[76]. K Southern hemisphere solar radiative force (orange)[95]. Yellow shadings represent periods of human occupation corresponding to the Lapita and Erueti phases.

proposed by Petchey et al.[22]. Increases in TOC and TN at 2850 BP signify higher aquatic productivity (Fig. 3), which is often indicative of human activities[60]. This is associated with a slight increase in Fe and Ti (Fig. 2) and in sediment grain size (Fig. 3), indicating a relative increase in erosion[61]. These sedimentary signatures also characterise subsequent periods of human presence and could be associated with land clearing, shifting cultivation and other horticultural practices[25]. Indeed, the highest peak in palmitone, the biomarker proxy for taro, the main contemporary staple crop of the region, coincides with this first period of human occupation (Fig. 2). Such a strong signal can be the result of early extensive and intensive taro cultivation, and possibly from a direct establishment of taro gardens on the shore of Emaotfer swamp. Previous studies of the Lapita diet based on the stable isotope analysis of bone collagen found a broad spectrum of both marine and terrestrial resources[62,63]. However, these analyses tend to over-represent the protein portion of the diet and underrepresent plant foods, therefore masking the horticultural component. Nevertheless, the early introduction of taro as part of a "transported landscape" has long been hypothesised for the Pacific Islands[64]. Previous linguistic and archaeological studies have associated the introduction of taro with the first settlements[43]. In our record, the coincidence of the first peak in palmitone and in faecal biomarkers explicitly signal the introduction and cultivation of taro by the first Lapita settlers.

A second period of human occupation is evident in the core with a second peak in palmitone and faecal biomarkers between 2400–2602 BP and 2298–2477 BP and follows after ca. 250 years of what could be a demographic decrease or even an absence of humans. Archaeological[54], anthropological2[65] and ancient DNA studies have revealed two distinct phases of human occupation of the Teouma site[3,66]. The biomarker record supports a subsequent separate settlement, which coincides with the Erueti period, identified by a 50 cm midden deposit that covered the Lapita cemetery at the Teouma site at ca. 2400 BP[54]. The Erueti period is characterised by substantial changes in pottery style, mortuary practice, and in diet[54,62,63].

After 2300 BP no further archaeological traces of human occupation were found at the Teouma site, which appears to have been abandoned until the development of a coconut plantation in the early 20th century[54]. However, in such a geodynamic context (sea level, uplift, earthquakes, etc.) signs of human occupation might be lost[67]. In our record, there are signs of a pause in human occupation after 2300 BP but an increase in faecal biomarkers and palmitone is evident around 2000 BP, a time devoid of archaeological traces at the Teouma site[54]. The swamp sediment is likely incorporating material from a larger catchment area than what is preserved at the archaeological site. The biomarkers related to human presence (coprostanol and epicoprostanol) and activity (palmitone) remain high from 2000 BP on, with some variations, indicating continuous human occupation and associated activities, including the presence of pigs and taro horticulture, on the island. The Mangaasi archaeological site northwest of Efate (Fig. 1) attests the continued presence of humans on Efate at ca. 2500–1200 BP[68]. During the last millennium, a demographic increase is evident from the widespread landscape features associated with cultivation and settlements that were observed by LIDAR imaging[56], which are coherent with the highest peaks in faecal biomarkers ca. 1000 BP observed in the core. This demographic increase could be associated with the Roi Mata's Domain, considered a period of unity and prosperity on the island[69] which is traced archaeologically in the small near-shore island of Artok northwest of Efate (Fig. 1).

**Late Holocene SPCZ shifts**
The hydrogen isotopic composition ($\delta^2$H) of *n*-alkanoic acids was measured in 91 samples to identify past changes in precipitation[49] (Fig. 4), which were carefully considered against the potential role for changing vegetation sources and the amount effect (Supplementary text S1). First molecular proxies for human activity were detected in the middle of a climatic shift towards considerably wetter conditions, which started ca. 2781–2964 BP. The abrupt increase in precipitation is part of a general pattern observed in the core, where stepwise shifts towards wetter conditions (10–20 ‰) were

recorded at several intervals between 3600 BP and 2200 BP, just before the appearance of the peat at 2000 BP (Fig. 4). The general trend towards wetter conditions was interrupted by drier and stable periods lasting a few centuries except for the period between 2600 and 2200 BP, which is characterised by greater fluctuations.

Changes in precipitation on the island of Efate are heavily influenced by the SPCZ, which is in turn closely linked to both ITCZ and ENSO dynamics. Past changes in the SPCZ position remain poorly constrained[32,70]. However, an increase in precipitation lasting approximately 200 years between 2700–2200 BP inferred from $^2$H-depleted C26 fatty acid in a sediment core from Sāmoa was associated with SPCZ expansion and a negative phase of the Interdecadal Pacific Oscillation[71]. An abrupt increase in sea surface temperature (SST) in the WPWP ca. 2800 BP could also have caused the wetter shift registered in our core[72], as higher SSTs are associated with increased precipitation from the SPCZ[73]. Model results suggest a southward and more variable position of the SPCZ in this period[34,71].

The SPCZ is a prominent extension of the ITCZ, yet how the SPCZ responded to changes in the ITCZ is not clear[32,74]. By comparing our climatic reconstruction of the SPCZ with ITCZ reconstructions, we could improve our understanding of the interaction between these climatic features. Paleoclimatic reconstructions available across the Pacific (Fig. 4) indicate a southward migration of the ITCZ over the late Holocene[75–77]. A steady decrease in Ti% influx in the Cariaco Basin indicates a southward shift of the ITCZ around 2850 BP[75]. If this shift was zonally symmetric, it could have had a role in the wetter trend recorded in Emaotfer core at that time. Seasonal increases in precipitation associated with the southward migration of the ITCZ are also recorded in the clay content of El Junco, in the Galápagos Islands[77]. Taking dating uncertainties into account, the two-step increase in precipitation observed at ~3200 BP and ~2000 BP in El Junco is also visible in Emaotfer record (Fig. 4). In the Western Pacific, a decrease in the Asian Summer Monsoon recorded in the $\delta^{18}$O of stalagmites from the Dongge Cave of Southern China at ~2800 BP is also connected with the southward movement of the ITCZ[76].

ENSO influences the location of the SPCZ, although this dynamic is complex and can determine regional differences not ultimately linked with the various ENSO flavours[78,79]. An increase in ENSO frequency ~3000 BP is observed across the tropical Pacific[80] and an associated drying trend starting at this time has been defined using carbon isotope ratios of leaves preserved in lake sediments from Western Australia, with dry events associated with prolonged extreme El Niño events between 2600 and 2000 BP[81]. These drying trends are coherent with $\delta^{18}$O measured in coral records from Papua New Guinea[82]. Increased frequency and amplitude of El Niño events between 2800 and 1500 BP are also recorded in the Galápagos[77,83]. Direct one-to-one ENSO reconstructions require yearly resolved records. However, intensified El Niño activity would have a cumulative drying effect on Efate that could be captured in our proxy records, and we do not observe such changes starting at 3000 BP. Wirrmann et al.[36] report a change from rainforest to grass between 2800 and 2400 BP at Emaotfer swamp. They associate these vegetation changes to an intensification of ENSO but lower resolution and age uncertainties limit the use of their record for comparison. Modern ENSO dynamics might only start exerting a role on Efate ca. 2600 BP when increased $\delta^2$H values indicate drier and more variable conditions, coincident with the extreme drying events recorded in Western Australia and Papua New Guinea[81,82].

**Discussion**
Our findings provide molecular evidence of first human settlements and the introduction of horticulture in Remote Oceania while revealing the climatic context at the time. The 5000-year-long sediment core from Emaotfer swamp generally agrees with the archaeological record of the Teouma site, and the high-resolution age constraints refine previous estimates of earliest possible use of the site. Our sediment core tracks the first human arrival at 2739–2879 BP, recorded as a peak in faecal biomarkers, coincident with the establishment of horticulture signalled by the introduction of taro, recorded by increased palmitone accumulation.

First settlers arrived during a time of hydroclimatic change from a drier period to a wetter one, most likely linked with a southward shift of the ITCZ and expansion of the SPCZ. The hydrology of Emaotfer Swamp was altered either as a response to an uplift event, the expansion of the SPCZ or to concomitant human activity. Whether humans, climate or tectonics caused these hydrological changes remains an open question. However, the growing evidence for the substantial alteration of the landscape seen by LIDAR imagery strongly favours a role for human activity and the continuity of culture on Efate[55]. Regardless, the sedimentary record highlights the influence of climate on human settlements in Remote Oceania. Climatic influences on human migrations and settlements have been described worldwide[84–89]. In the Pacific, climate has been suggested as a factor influencing navigation[11,12], horticultural expansion and as a possible explanation for the dietary differences observed between subsequent human groups, as the difference observed between the Lapita and the subsequent Erueti people[90].

The Emaotfer climatic record provides further context to the differences characterising the two archaeological phases of occupation at the Teouma site. Lower $\delta^2H$ values in the leaf wax record, indicative of an increasingly wet climatic window from 2781–2964 BP until 2602–2713 BP, would have provided the ideal conditions for the establishment of horticulture during the Lapita period, as indicated by the highest concentration of palmitone, indicative of taro cultivation. The establishment and expansion of horticulture during the Lapita period would have relied on these favourable climatic conditions; the availability of water, along with fertile soils and cultivation management practices, could have contributed to successful horticultural establishment. Following previous reconstructions of climate windows for human migration in the Pacific[11,15], our climatic reconstruction of Emaotfer raises several questions. The drier period preceding the first human signal in the core may have played a role in triggering human migrations in search for new favourable settings or the end of the wetter period characterising their settlements could have contributed to site abandonment. The coeval Lapita settlement of Tonga[23] could indicate that similar drivers influenced the Lapita expansion further to the east.

The brief Lapita period began and ended during a climate transition period towards wetter conditions, which favoured horticultural development. A second peak in palmitone and faecal biomarkers at 2400 BP corresponds with the Erueti phase observed in the archaeological records. The more variable climate during the Erueti period was likely influenced by ENSO effects on the SPCZ. The more variable conditions associated with the Erueti period may have contributed to a decline in human presence, as indicated by an abrupt decrease in the archaeological signal. The challenges posed by an unstable climate could have influenced the decision to abandon or reduce occupation in certain areas. The lower palmitone signal during the Erueti period might also reflect the impact of such variable climatic conditions on crop yields, or that dryland horticulture was prioritised. The changes in precipitation patterns could have affected agricultural productivity, potentially leading to food scarcity or agricultural difficulties for the Erueti settlement. Despite some fluctuations in faecal biomarkers and palmitone concentrations possibly related to demographic changes, these sedimentary biomarkers reveal that humans have had a sustained presence in the catchment.

Overall, the biomarkers preserved in the Emaotfer sediment provide evidence of first horticultural activities and human presence in Remote Oceania and demonstrate the importance of precipitation variability in determining both when humans settled remote islands, as well as the resource-acquisition strategies they employed upon arrival. These findings advance our understanding of human history in the Pacific and provide insights into the influence of climate on human societies.

## Materials and methods
### Study site
Emaotfer is a shallow swamp located on a limestone terrace 3 m above sea level on the southern coast of the Island of Efate. The Emaotfer basin was likely formed when the catchment was inundated with marine waters prior to coastal uplift during the early Holocene. The swamp is located east of the Teouma Graben, into which the Teouma River flows (Fig. 1). The water depth is influenced by seasonal changes in groundwater and precipitation with a wet/cyclone season from November to April and a dry season from May to October. Annual rainfall is around 2100 mm but can vary strongly during El Niño (dry) and La Niña (wet) periods. Two palaeoecological studies have previously reconstructed vegetation changes from Emaotfer swamp, but have not detected definitive signals of human presence corresponding with the time of occupation of the Teouma archaeological site[36,37].

### Coring and sub-sampling
In July 2017, we cored Emaotfer swamp using multiple drives of a 50 cm length Russian peat corer (lat. 17°47'6.66" S, long. 168°23'55.22" E). Water depth at the coring site was 0.3 m. The 50 cm core sections were wrapped twice in plastic foil, placed in halved PVC tubes, stored in a cooler and flown back to the Eawag laboratories in Switzerland the following week. The deepest core section retrieved reached 425 cm in depth and was subsampled at 1 cm resolution following XRF scanning and samples were then freeze-dried. Macrofossils were separated for radiocarbon measurements. The sub-sample was then split with 1.5 cm$^3$ used for bulk analyses while the rest of the sediment was used for biomarker extraction (up to ~3 g of dried sediment).

### Chronology
Radiocarbon dating of 57 macrofossils from 48 unique depths (Table S1) was carried out at the Laboratory of Ion Beam Physics of ETH Zurich. Plant remains were chosen for the measurements. Most of the macrofossils samples are from unidentifiable leaf and root materials, principally from monocotyledons (e.g. Cyperaceae and Pandanaceae) (Fig. S5). We suggest that such materials are likely to be short-lived with limited in-built age potential. After an acid-base-acid chemical cleaning treatment[91] 41 samples underwent graphitization before measurement, while smaller samples were directly measured with an accelerator mass spectrometer[92]. Data evaluation and corrections were done following the procedures described in Welte et al. [93]. R Statistical Software (v4.2.0, R core team 2022) was used to perform all data analysis and visualization. The age-depth deposition model was performed with the package rbacon version 2.5.8[94]. Radiocarbon dates were calibrated with the Southern Hemisphere calibration curve shcal20[95]. Seventeen samples were excluded from the model. Of these, seven samples were identified as root intrusions and eight samples were excluded as their older ages and $\delta^{13}C$ values above −15‰ indicate that these samples integrated old carbon derived from the limestone catchment. Postbomb dates were not included in the age-depth model. All other dated samples, including the reversed age at 181 cm, were initially added to the model input. However, the final Bayesian age-depth model computed by the rbacon package excluded this sample as being too old for being consistent with the other calibrated ages and the deposition model.

### Bulk geochemical analyses
Bulk geochemical analyses were carried out in the Sedimentology laboratories at Eawag, Dübendorf. Downcore total carbon (TC) and total nitrogen (TN) content were measured with an EURO Elemental Analyser (EA) 3000 for a total of 104 samples. Total inorganic carbon (TIC) was measured with a titration Coulometer (CM5015). Total organic carbon (TOC) was calculated using the equation TOC = TC − TIC. The bulk sediment $\delta^{15}N$ was measured on an EA-IRMS (EA Vario Pyro Cube by Elementar and IRMS by GV Instruments, Isoprime). Elemental counts were measured using an Avaatech XRF core scanner with an Oxford 100 Watt X-ray source with Rhodium anode and Canberra X—Pips and Canberra DSA 1000 (MCA) detector. The sediment and peat core sections were carefully levelled and covered with a 4 µm thick ultralene plastic film. Two different settings were applied for the scan: 10 kV with 30 s count time, no filter for the lighter elements, and 30 kV with thin Pd filter and 30 s count time for the heavier elements. Step size was 5 mm.

## Biomarker analysis

Lipid extraction, purification and quantification were performed in the Sedimentology laboratories at Eawag, Dübendorf. Total lipid content was extracted from 112 sediment samples distributed along the core length. Sediment samples were extracted in a mixture of DCM/MeOH (9:1, v/v) with a Dionex ASE 350 (Thermo Scientific). Lipid saponification and column chromatography of the neutral fraction and derivatization of the sterols were performed as in Krentscher et al.[47]. The acid fraction was separated, methylated and purified as in Ladd et al.[49].

Compounds were identified and quantified by gas chromatography–mass spectrometry (GC–MS) as described in Krentscher et al.[47]. Ketones and the derivatized sterols were run with a Selected Ion Monitoring method targeting ions (Supplementary text S3). An external standard with the targeted compounds was added for identification and quantification via external calibration curve. The ratio between the sum of coprostanol and epicoprostanol and cholestanol was included in the results (Fig. S6).

The hydrogen isotopic analyses of leaf waxes were performed in the Stable Isotope Ecology Lab at the University of Basel. The samples were analysed on a Trace GC Ultra gas chromatograph (GC) coupled to a Thermo Delta V Plus isotope ratio mass spectrometer (IRMS) via a GC Isolink operated in pyrolysis mode and ConFlo IV interface (Thermo Fisher Scientific, Bremen, Germany). The measured values were normalised to the VSMOW/SLAP scale (Vienna Standard Mean Ocean Water/Standard Light Antarctic Precipitation) using hydrogen isotope standards purchased from Arndt Schimmelmann at Indiana University. Measurement accuracy and precision were assessed from a quality control standard, a hydrocarbon fraction from oak leaves that were originally collected in Berkeley, California. The average $\delta^2H$ value of the $n$-C29 alkane in this standard is $-142.4 \pm 3.7$ ‰ ($n = 868$, going back to 2014). The standard was analysed 53 times with our sample set, with a mean $\delta^2H$ value of $-141.4$‰, which is offset from the calibrated value by 1.2 ‰. The standard deviation of these analyses was 1.6 ‰. Additional details on the robustness of the $\delta^2H$ signal can be found in the Supplementary text S1 and Figs. S7–S10.

## Reporting summary

Further information on research design is available in the Nature Portfolio Reporting Summary linked to this article.

## Data availability

Supplementary Materials are available for this paper. All data needed to evaluate the conclusions in the paper are present in the paper and the Supplementary Materials. The dataset underlying the study is deposited in the ETH Zurich research collection at https://doi.org/10.3929/ethz-b-000652386.

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

## Acknowledgements

The research permit was approved by the Vanuatu National Cultural Council and the Department of Environmental Protection and Conservation (DEPC). We thank Richard Shing and Henline Mala from the Vanuatu Cultural Center and Reedly Tari, Donna Kalfatakmoli and Primrose Malosu from DEPC for their guidance in the permit process. Danny Nef assisted with 2017 fieldwork. Irene Brunner from the sedimentology group at Eawag conducted bulk analysis, Caroline Welte, Silvia Bollhalder, and Karin Wyss Heeb from the Ion Beam Physics department of ETH Zurich, and Anita Schlatter from the sedimentology group, Eawag, helped with the radiocarbon dating. We thank the Teouma site leaseholder M. Robert Monvoisin for granting permission to access and core in their land and Stuart Bedford for support in the field as well as for constructive feedback. We thank Charmaine Bassfeld, Shannon Dyer, Erik Hegenberg, Gioele Scacco, and Lucas Soliva for technical support. We thank Gabriele Consoli, Irka Hajdas, Dave Jansen, Benjamin Keenan, Nannan Li, and Tobias Schneider for their constructive feedback during the preparation of this manuscript. We would like to thank Rebecca Kinaston for providing a correction on the preprint version of the manuscript. We would like to thank three anonymous reviewers whose comments have helped improve the quality of this paper. This work is part of the Swiss National Science Foundation (SNSF) funded MACRO project

(Grant Nr. PP00P2_163782 to ND). Additional laboratory work was funded by the Tailwind grant of Eawag Switzerland to GC.

## Author contributions

N.D. and M.P. conceived the project. N.D. acquired the funding for the project. S.N.L. and G.C. contributed to the design of the study. N.D. and G.C. coordinated the study. N.D., M.P., S.N.L., R.L. and G.C. conducted the fieldwork. C.K., E.A., S.N.L., R.L., N.D. and G.C. conducted the lipid biomarker quantification and interpretation. D.B.N. and S.N.L. performed the compound-specific stable isotope analyses. R.L. and G.C. performed the XRF core scanning and grain size analysis. N.D. directed the bulk geo-chemical analyses. G.C. created the figures and led the writing of the paper to which all authors contributed. All authors discussed the results and commented on the manuscript.

## Competing interests

The authors declare no competing interests.
