## [Transparent Peer Review file · Communications Earth & Environment]

Sedimentary biomarkers of human presence and taro cultivation reveal early horticulture in Remote Oceania

Corresponding Author: Dr Giorgia Camperio

Version 0:

Decision Letter:

Dear Ms Camperio,

Please allow us to sincerely apologise for the long delay in sending a decision on your manuscript titled "Wetter climate favouring early Lapita horticulture in Remote Oceania". It has now been seen by 3 reviewers, and we include their comments at the end of this message. They find your work of interest, but some important points are raised. We are interested in the possibility of publishing your study in Communications Earth & Environment, but would like to consider your responses to these concerns and assess a revised manuscript before we make a final decision on publication.

We therefore invite you to revise and resubmit your manuscript, along with a point-by-point response that takes into account the points raised. Please highlight all changes in the manuscript text file.

Please use the following link to submit your revised manuscript, point-by-point response to the referees' comments (which should be in a separate document to any cover letter), a tracked-changes version of the manuscript (as a PDF file) and the completed checklist:

Link Redacted

We hope to receive your revised paper within six weeks; please let us know if you aren't able to submit it within this time so that we can discuss how best to proceed. If we don't hear from you, and the revision process takes significantly longer, we may close your file. In this event, we will still be happy to reconsider your paper at a later date, as long as nothing similar has been accepted for publication at Communications Earth & Environment or published elsewhere in the meantime.

Please do not hesitate to contact us if you have any questions or would like to discuss these revisions further. We look forward to seeing the revised manuscript and thank you for the opportunity to review your work.

Best regards,

Yiming Wang, PhD
Editorial Board Member
Communications Earth & Environment
orcid.org/0000-0003-3228-5592

Joe Aslin
Deputy Editor
Communications Earth & Environment

EDITORIAL POLICIES AND FORMATTING

We ask that you ensure your manuscript complies with our editorial policies. Please ensure that the following formatting requirements are met, and any checklist relevant to your research is completed and uploaded as a Related Manuscript file

type with the revised article.

Editorial Policy: [Policy requirements](https://www.nature.com/documents/nr-editorial-policy-checklist.pdf) (Download the link to your computer as a PDF.)

- Behavioural and social science
- Ecological, evolutionary & environmental sciences
- Life sciences

<https://www.nature.com/documents/nr-reporting-summary.zip>

Furthermore, please align your manuscript with our format requirements, which are summarized on the following checklist: [Communications Earth & Environment formatting checklist](https://www.nature.com/documents/commsj-phys-style-formatting-checklist-article.pdf)

and also in our style and formatting guide [Communications Earth & Environment formatting guide](https://www.nature.com/documents/commsj-phys-style-formatting-guide-accept.pdf) .

*** DATA: Communications Earth & Environment endorses the principles of the Enabling FAIR data project (<http://www.copdess.org/enabling-fair-data-project/>). We ask authors to make the data that support their conclusions available in permanent, publically accessible data repositories. (Please contact the editor if you are unable to make your data available).

All Communications Earth & Environment manuscripts must include a section titled "Data Availability" at the end of the Methods section or main text (if no Methods). More information on this policy, is available at <http://www.nature.com/authors/policies/data/data-availability-statements-data-citations.pdf>.

If a community resource is unavailable, data can be submitted to generalist repositories such as [figshare](https://figshare.com/) or [Dryad Digital Repository](http://datadryad.org/). Please provide a unique identifier for the data (for example a DOI or a permanent URL) in the data availability statement, if possible. If the repository does not provide identifiers, we encourage authors to supply the search terms that will return the data. For data that have been obtained from publically available sources, please provide a URL and the specific data product name in the data availability statement. Data with a DOI should be further cited in the methods reference section.

REVIEWER COMMENTS:

Reviewer #1 (Remarks to the Author):

The manuscript by Camperio et al. presents a data-based assessment of human history in the remote Pacific using the organic geochemistry of sedimentary lipids. While the expansion of humans across the Pacific region is a well-studied topic, the authors provide compelling high-resolution datasets from an important study site in Vanuatu. Although results from only one site are presented in this manuscript, the outstanding contribution of this work is the high quality and high temporal resolution of their analysis from this carefully chosen location. The authors' methods description of the advanced molecular biomarker approach and other techniques are clear in the main and supporting documents. I have three main comments on the manuscript and several smaller comments that follow.

The first main comment relates to the interpretation of the $\delta^2\text{H}$ values. A main message of the paper, reflected in the first word of the title, relates to the interpretation of depleted lipid deuterium values as showing a wetter climate condition. In the body of the main paper, the authors state that the hydrogen isotope composition of leaf waxes 'reflect changes in precipitation' (line 104) and cite Sachse et al. 2012 in support of this statement. This is a bold oversimplification of the

mechanisms affecting d2H in sedimentary lipids, and the paper by Sachse et al. is essentially a cautionary tale of the dangers of over-interpreting such data. I don't think the authors are wrong necessarily in their interpretation, and they do a good job in the supporting document (supplementary lines 98-154) of detailing the potential effects of post-rainfall evaporation/evapotranspiration and changes in plant communities on biosynthesis processes. But missing is discussion of 'changes in the source and transport pathways of the rain/clouds' (supplementary line 111). These 'source effects' has been shown to be a dominant control, greater than amount effects, on other Pacific Islands (e.g. Scholl et al. 1996 doi: 10.1029/95wr02837). Also missing is inclusion of evidence for the amount effect. The authors state that the "amount effect" describes higher precipitation d2H values with decreasing rainfall' and cite seminal, but mostly theoretical, work from the 60s in support. Missing also from the supplementary discussion is proving the importance of the 'amount effect' with recent data of rainfall and d2H values in the region and transparency about just how much rainfall variation is accountable by this effect. Including additional, even if brief, information on both the source effect in the ITCZ region, and further justification for the amount effect interpretation would strengthen their interpretation.

The second main comment relates to the radiocarbon data and chronologies. I think the effort the authors invested in their radiocarbon measurements and chronology is commendable; it is great to see so many dates. But in order to have 'all data needed to evaluate the conclusions in the paper' (line 422) and to better understand the 17 of 57 (nearly one-third) measurements excluded from the chronologies, more information would be helpful regarding the material dated. Table S1 lists 'plant remain' as the material for 36 of the 57 measurements. It would be valuable to know more about these plant remains. The authors are experts in the identification of macrofossil remains in tropical sedimentary settings, so please provide more botanical/taxonomic information for the plant remains so that readers can better evaluate these data and the author's conclusions. For example, it is stated that some were identified as 'root intrusions' (line 372) suggesting that more information is available. Also, of the chronology-excluded measurements, no discussion, or even mention (lines 371-375), is given to the strikingly old sample of plant remains at 181cm (Table S1) that sits directly at the level of increased faecal sterols and palmitone that is one of the main messages of the paper. I would agree that this measurement does not belong in the chronology, but it also seems that this might be anthropogenic and indicative of land use during the early Lapita period. Because there is similar aged material from deeper in this same core (and with similar d13C value, i.e. 351 cm) there is likely old, preserved material elsewhere in the swamp that could have been disturbed/excavated - I wonder if this can be seen as evidence of horticulture practices and wet taro cultivation and 'establishment of taro gardens on the shore of Emaotfer' (Line 190). Perhaps some brief discussion in the supplementary document (add a chronology section and expand on lines 62-64) or to the methods Chronology section is warranted for this interesting anomaly right in the biomarker hotspot and in the middle of the section where the authors argue the chronology is most reliable (Figure S4). These deep reversals were not encountered in previous study of the same location from the same swamp (ref 80) which also merits some mention/discussion.

The final main comment is about the timing of first rise in faecal sterols and palmitone relative to the wet-climate proxy. Taking the d2H interpretation at face value – that decreasing C30 d2H values indicate wetter climate conditions – I agree with the author's statement that 'first settlers arrived during a time of hydroclimatic change' (line 288). But the brief Lapita period (as evidenced by the biomarkers) comes and goes during the period of transition in d2H value, and these biomarker indicators are long gone by the time the d2H reaches its protracted low values around 2.75ka. This timing makes it hard not to flag the statement 'the end of the brief wetter period, which favoured horticultural development, marks the end of the Lapita settlement at the site' (lines 316-317) because, according to the biomarker indicators, the Lapita period both began and ended during the transition period and was absent from the most depleted d2H (assumed wet) period. The colored bands that the authors draw in Figure 2 for 'Lapita' and 'Erueti' in fact seem to have similar d2H values, which is in contrast to the statement 'the more variable and drier climate during the Erueti period' (line 319). While I don't have a problem with the main message of the manuscripts title, it seems to me it is hard to link the end of the Lapita settlement with a change to drier conditions (lines 316-317).

Line 2. 'favouring' vs 'favoured' – perhaps past tense best here.

Line 32. The term 'quasi-simultaneously' seems unnecessarily hedging and could be stated more strongly, since the biomarker increases seem to me (eyeballing Figures 2 and S5) that they are occurring in the same levels. Or include some kind of quantitative estimate of 'within x years' to strengthen this result to be more satisfying than 'quasi-simultaneous'.

Line 34. Consider 'wetting' or 'increasing wetness' rather than 'wetter' – see third main comment above.

Line 45. Should this be 'the islands of Southeast Asia and Papua'?

Line 51. I tripped on 'inducing' here – perhaps 'resulting in' or 'motivating'.

Lines 54-57. There are more than a few existing climate records, they are just not high quality by 2024 standards. So paucity is not the strongest selling point for this excellent work by the authors. Perhaps re-phrase this section to be more specific about the Pacific climate knowledge gaps filled by this manuscript, like resolution, chronology, multi-proxy, etc.

Lines 58-60. This sentence is dangling in space between paragraphs – does it belong with the next paragraph (lines 61-72)?

Lines 87-89. Any previous sediment work from Emaotfer should be cited somewhere around here, including the initial work by Wirmann whose deeper, older core seems like it was taken very close to the one presented in the manuscript. Excluding other sediment papers from Emaotfer here makes it sound like this is the first time it has been studied, which is not the case. Lines 117-119. Figure 1C is called out for observable beach ridges at 4m – it's not clear to me where these are in the figure. If this is important, consider improving the figure to make these geomorphological units clear.

Line 133. It would be useful to cite the appropriate figure for the C:N (Fig 4) and d15N values (Fig S1?).

Line 149. Is the 'drying trend preceding 2900 BP' truly a drying trend? Compared to later fluctuations it seems somewhere between stable and getting wetter over the previous millennium before 2.9ka (taking the interpretation of depleted C30 d2H shows more rain at face value). Perhaps rephrase.

Lines 167-170. This is a good argument for creating an independent chronology across the Lapita period, but totally glosses

over the big reversal at 181cm. See second main comment.

Line 173. I tripped over the ordering of dates and periods in this sentence. Perhaps it is better to stay consistent with earliest-to-latest order and change this to '2624-2750 BP to 2739-2879 BP'.

Line 174. Perhaps 'indeed' would be better as 'likely' here given the uncertainties in the ranges reported in this sentence.

Line 181. Consider change to '2870-2920 BP' for consistency as in Line 173 comment.

Lines 193-194. Well said.

Lines 197-199. I agree; see comment about Line 32.

Lines 209-226. This message, especially Lines 213-215, seems rather important. Perhaps this should be highlighted briefly in the abstract.

Lines 261. Perhaps 'manifestations' rather than 'flavours'.

Line 303. 'increasingly wet' rather than 'wetter'. See third main comment.

Lines 316-317. This statement is poorly supported by the data presented. The wet period, as interpreted by depleted d2H values, outlasts the palmitone and faecal peak, suggested the end of the Lapita settlement during the transition to the wetter conditions, and gone before the wettest.

Line 319. Regarding the 'more variable and drier climate during the Erueti period', I see the argument for more variable. But would say that the wetness proxy has pretty much the same value for both human periods.

Lines 363-364. Please detail these plant remains. See second main comment.

Lines 371-375. The details of why these measurements were excluded is useful to evaluate the chronology. But the deeply reversed date right initial human settlement (181 cm, d13C = -28) is glaringly absent from these justifications. See second main comment.

Lines 379-381. It would be interesting to see these TIC values somewhere, given the comments on watershed erosion in the manuscript (e.g. Lines 157-158). Perhaps added to Fig 4D, or an expanded Fig. S1.

Line 716-717. Please cite a source for the Lapita archeological sites mapped.

Lines 719-721. Please cite a source for the geological data mapped.

Lines 758-759. If the wet period shaded in blue is based on the author's interpretation of the d2H values, then the darker shading should extend to the period of most depleted d2H to around 2.7ka.

Supplementary Information file

Figure S1. The data plotted do not match the caption. The interpretation of 'marine influence' seems to have the arrow going the wrong direction, since the text argues that higher d15N values reflect oceanic nitrogen sources. It is interesting to note that some of the peat has marine-like d15N. There is a period missing at the end of the caption.

Lines S159-S160. These descriptions of the panels do not match the data presented.

Lines S160-S162. I can see where the authors are going with this, but the colors are just a ramp of depths. This interpretation is stronger based on the crossplot groupings, and it may be more compelling if the cluster of bottom left points were labeled as 'lacustrine' or 'lake' to reflect this interpretation and to be consistent with Figure S2. This sentence could be rephrased to be based on the separations rather than the depths/colors.

Lines S201-S202. Should this title refer to Figure S6 rather than Figure S7?

Table S1. That is a lot of decimal places for 14C and 13C measurements. Convention is to report 14C years and SD to the nearest year at most. It would be useful to have the postbomb dates also reported as F14C as per convention. One decimal place for d13C is more reflective of accuracy, and what most readers are used to seeing, consider changing. Please provide taxonomic information for 'plant remains' and 'leaves' and 'seeds', see second main comment. The caption mentions that samples with roots are indicated with an asterisk, but none are provided; please provide.

Reviewer #3 (Remarks to the Author):

The manuscript by Camperio et al. provides a novel and interesting study of Lapita agriculture and environmental change via the analysis of faecal sterols and palmitone, which is a biomarker for ingested/digested taro. The samples were obtained from a sediment core and coupled with a robust series of radiocarbon dates, elemental and isotopic analyses of sediments, age-depth modeling, and analyses of alkanolic acids, all of which provide detailed symphony of data that reveal 6000 years of Vanuatu's past. I commend the authors for the design of the research itself, which has much to offer in terms of connecting questions about agriculture and human presence with climate variability. The paper is very well written and sourced, with diligent attention to the presentation of the data in the figures and supplementary tables. I think as a whole the manuscript argues a very strong case for the production and consumption of taro in Vanuatu during the colonization phase of Lapita, and also provides further information about the logistics of that colonization with the related climate and geological data.

The figures are particularly compelling-- I studied them and am satisfied with how they are used to depict the results of the analyses. Overall the structure of the manuscript is well organized, although I found it odd to have the materials and methods follow after the discussion.

I think this is a groundbreaking manuscript. I suggest the authors create a title that reflects the analysis of faecal sterols and palmitone, and their use as indicators for taro consumption. Although the results indicating wetter environments at the time of Lapita is important and robustly supported by the analyses, there have been many climate-related Lapita papers over the decades, but none about the molecular data and analyses described here. The manuscript truly stands out--- it deserves a more distinctive and representative title.

Reviewer #4 (Remarks to the Author):

This manuscript presents sedimentological, geochemical, and biomolecular data to assess the timing of human arrival and early horticultural activity on Efate and their relationship with climatic events. I find the paper interesting and an important advancement in the application of biomarkers in archaeological and ecological projects. While I find no issues with the paper's methods and data presentation, I include here several comments regarding data interpretation and discussion. This is strictly for the authors' consideration in the hopes of strengthening the paper and I advocate for its publication regardless of the incorporation of these comments.

Comment 1: Highlighting the importance of the Emaotfer fecal stanol record. In my opinion, the stanol data shown here help clarify the difference between nonhuman and human coprostanol signals. The supplemental text gets into this somewhat, but I suggest adding a few sentences to the main text starting on line 171 that acknowledge A) the low level coprostanol values before humans arrive, B) how this demonstrates nonhuman coprostanol production, and C) how this study is uniquely situated to differentiate nonhuman vs. human-related coprostanol production because of the relatively recent and abrupt arrival of humans here that is difficult to find in other areas of the world. I look forward to citing this paper for this very reason and I recommend calling out this aspect of the study in the main text.

Comment 2: Catchment area: In line 215 the authors write, "The swamp sediment is likely incorporating material from a larger catchment area than what is preserved at the archaeological site." Have the authors tried modelling the current catchment area with GIS software? Incorporating a watershed boundary line into Figure 1 might improve the paper.

Comment 3: Impact of pigs: In line 217, the authors write, "The biomarkers related to human presence and activity remain high from 2000 BP on, with some variations, indicating continuous human occupation of the island." How much do nonhuman domesticates, like pigs, have to do with the post-2000 BP disconnect between high coprostanol values and low local archaeological data? Did pigs become feral on Efate? While pigs are discussed in the supplemental text, I believe having at least a sentence that acknowledges potential coprostanol input from nonhuman domesticates is warranted in the main text.

Communications Earth & Environment is committed to improving transparency in authorship. As part of our efforts in this direction, we are now requesting that all authors identified as 'corresponding author' create and link their Open Researcher and Contributor Identifier (ORCID) with their account on the Manuscript Tracking System prior to acceptance. ORCID helps the scientific community achieve unambiguous attribution of all scholarly contributions. You can create and link your ORCID from the home page of the Manuscript Tracking System by clicking on 'Modify my Springer Nature account' and following the instructions in the link below. Please also inform all co-authors that they can add their ORCIDs to their accounts and that they must do so prior to acceptance.
<https://www.springernature.com/gp/researchers/orcid/orcid-for-nature-research>

Author Rebuttal letter: The author's response to these comments can be found at the end of this file.

Version 1:

Decision Letter:

Dear Ms Camperio,

Your manuscript titled "Sedimentary biomarkers of human presence and taro cultivation reveal early horticulture in Remote Oceania" has now been seen by our reviewers, whose comments appear below. In light of their advice we are delighted to say that we are happy, in principle, to publish a suitably revised version in Communications Earth & Environment.

We therefore invite you to edit your manuscript to comply with our format requirements and to maximise the accessibility and therefore the impact of your work.

EDITORIAL REQUESTS:

*****Please take care to match our formatting and policy requirements. We will check revised manuscript and return manuscripts that do not comply. Such requests will lead to delays. *****

SUBMISSION INFORMATION:

OPEN ACCESS:

Communications Earth & Environment is a fully open access journal. Articles are made freely accessible on publication. For further information about article processing charges, open access funding, and advice and support from Nature Research, please visit <https://www.nature.com/commsenv/open-access>

Link Redacted

Best regards,

Yiming Wang, PhD
Editorial Board Member
Communications Earth & Environment
orcid.org/0000-0003-3228-5592

Joe Aslin
Deputy Editor,
Communications Earth & Environment
<https://www.nature.com/commsenv/>
Twitter: @CommsEarth

REVIEWERS' COMMENTS:

Reviewer #1 (Remarks to the Author):

Second review of paper 'Sedimentary biomarkers of human presence and taro cultivation reveal early horticulture in Remote Oceania' by Camperio et al.

This manuscript is improved significantly based on the author's responses to my initial comments, and those of the other reviewers. I thank the authors for their open-mindedness to the initial comments and for their thoughtful consideration. The quality, accuracy, and transparency of the presented work are all improved.

I have no other comments on this manuscript.

Reviewer #3 (Remarks to the Author):

I approve of the revisions.

I see that this journal is using a transparent peer review model. In that case, could you please spell my name correctly? It is Julie Field (no "s" at the end). It was entered incorrectly into your system by someone else. Thank you!

Reviewer #4 (Remarks to the Author):

I believe the manuscript is ready for publication. Congratulations to the authors for this contribution.

Florence, 8 July 2024

Dear Reviewers,

Thank you for your thoughtful and constructive feedback on our manuscript, COMMSENV-24-0536-T, and for the time and effort you have invested in reviewing our work. We have carefully considered all the comments and suggestions and have made the necessary revisions, which we believe have greatly improved the original manuscript.

Below, we provide a detailed response to each comment.

Sincerely,

Giorgia Camperio

Eawag - Department of Surface Waters - Research & Management

Überlandstrasse 133, 8600, Dübendorf, CH

giorgia.camperio@eawag.ch

REVIEWER COMMENTS:

Reviewer #1 (Remarks to the Author):

The manuscript by Camperio et al. presents a data-based assessment of human history in the remote Pacific using the organic geochemistry of sedimentary lipids. While the expansion of humans across the Pacific region is a well-studied topic, the authors provide compelling high-resolution datasets from an important study site in Vanuatu. Although results from only one site are presented in this manuscript, the outstanding contribution of this work is the high quality and high temporal resolution of their analysis from this carefully chosen location. The authors' methods description of the advanced molecular biomarker approach and other techniques are clear in the main and supporting documents. I have three main comments on the manuscript and several smaller comments that follow.

The first main comment relates to the interpretation of the $\delta^2\text{H}$ values. A main message of the paper, reflected in the first word of the title, relates to the interpretation of depleted lipid deuterium values as showing a wetter climate condition. In the body of the main paper, the authors state that the hydrogen isotope composition of leaf waxes 'reflect changes in precipitation' (line 104) and cite Sachse et al. 2012 in support of this statement. This is a bold oversimplification of the mechanisms affecting $\delta^2\text{H}$ in sedimentary lipids, and the paper by Sachse et al. is essentially a cautionary tale of the dangers of over-interpreting such data. I don't think the authors are wrong necessarily in their interpretation, and they do a good job in the supporting document (supplementary lines 98-154) of detailing the potential effects of post-rainfall evaporation/evapotranspiration and changes in plant communities on biosynthesis processes.

But missing is discussion of 'changes in the source and transport pathways of the rain/clouds' (supplementary line 111). These 'source effects' has been shown to be a dominant control, greater than amount effects, on other Pacific Islands (e.g. Scholl et al. 1996 doi: 10.1029/95wr02837). Also missing is inclusion of evidence for the amount effect. The authors state that the "amount effect" describes higher precipitation $\delta^2\text{H}$ values with decreasing rainfall' and cite seminal, but mostly theoretical, work from the 60s in support.

Missing also from the supplementary discussion is proving the importance of the 'amount effect' with recent data of rainfall and $\delta^2\text{H}$ values in the region and transparency about just how much rainfall variation is accountable by this effect. Including additional, even if brief,

information on both the source effect in the ITCZ region, and further justification for the amount effect interpretation would strengthen their interpretation.

⇒ Thank you for raising these issues. We have modified our main text on lines 111-117 to make it clear that we have considered them and we discuss them in more detail in the supplementary text 3 (S104-S186). In the supplementary text 3 we expanded our discussion on the role of the “amount effect” and have now included the suggested reference and additional references about the mechanisms likely underlying the empirically observed amount effect. In particular, we have added references to studies by Risi et al. (2008) and Conroy et al. (2013), which provide theoretical, empirical, and modeling-based evidence for the amount effect at low-elevation maritime tropical Pacific sites, like for the low-elevation coastal site in our study.

The second main comment relates to the radiocarbon data and chronologies. I think the effort the authors invested in their radiocarbon measurements and chronology is commendable; it is great to see so many dates. But in order to have ‘all data needed to evaluate the conclusions in the paper’ (line 422) and to better understand the 17 of 57 (nearly one-third) measurements excluded from the chronologies, more information would be helpful regarding the material dated. Table S1 lists ‘plant remain’ as the material for 36 of the 57 measurements. It would be valuable to know more about these plant remains. The authors are experts in the identification of macrofossil remains in tropical sedimentary settings, so please provide more botanical/taxonomic information for the plant remains so that readers can better evaluate these data and the author’s conclusions. For example, it is stated that some were identified as ‘root intrusions’ (line 372) suggesting that more information is available.

⇒ We have added pictures of the remains which were radiocarbon dated in the Supplementary Information (fig S5). Most of the materials dated are from unidentifiable leaf and root materials, principally from monocotyledons (e.g. Cyperaceae and Pandanaceae). We suggest that such materials are likely to be short-lived with limited in-built age potential (line 401 - 405).

Also, of the chronology-excluded measurements, no discussion, or even mention (lines 371-375), is given to the strikingly old sample of plant remains at 181cm (Table S1) that sits directly at the level of increased faecal sterols and palmitone that is one of the main messages of the paper. I would agree that this measurement does not belong in the chronology, but it also seems that this might be anthropogenic and indicative of land use during the early Lapita period. Because there is similar aged material from deeper in this same core (and with similar $\delta^{13}\text{C}$ value, i.e. 351 cm) there is likely old, preserved material elsewhere in the swamp that could have been disturbed/excavated - I wonder if this can be seen as evidence of horticulture practices and wet taro cultivation and ‘establishment of taro gardens on the shore of Emaotfer’

(Line 190). Perhaps some brief discussion in the supplementary document (add a chronology section and expand on lines 62-64) or to the methods Chronology section is warranted for this interesting anomaly right in the biomarker hotspot and in the middle of the section where the authors argue the chronology is most reliable (Figure S4). These deep reversals were not encountered in previous study of the same location from the same swamp (ref 80) which also merits some mention/discussion.

⇒ We have added a short statement on lines 182-187, and 411-412 to acknowledge this reversal. We agree with the reviewer's suggestion of a possible anthropogenic driven soil erosion as the possible cause of the reversal. We also added more detailed information in the table explaining that the comparison with the record of Wirrmann et al. 2011 is limited by the lower resolution of dating of this record and the age uncertainties (189-192).

The final main comment is about the timing of first rise in faecal sterols and palmitone relative to the wet-climate proxy. Taking the $\delta^{2}H$ interpretation at face value – that decreasing $\delta^{2}H$ values indicate wetter climate conditions – I agree with the author's statement that 'first settlers arrived during a time of hydroclimatic change' (line 288). But the brief Lapita period (as evidenced by the biomarkers) comes and goes during the period of transition in $\delta^{2}H$ value, and these biomarker indicators are long gone by the time the $\delta^{2}H$ reaches its protracted low values around 2.75ka. This timing makes it hard not to flag the statement 'the end of the brief wetter period, which favoured horticultural development, marks the end of the Lapita settlement at the site' (lines 316-317) because, according to the biomarker indicators, the Lapita period both began and ended during the transition period and was absent from the most depleted $\delta^{2}H$ (assumed wet) period. The colored bands that the authors draw in Figure 2 for 'Lapita' and 'Erueti' in fact seem to have similar $\delta^{2}H$ values, which is in contrast to the statement 'the more variable and drier climate during the Erueti period' (line 319). While I don't have a problem with the main message of the manuscript's title, it seems to me it is hard to link the end of the Lapita settlement with a change to drier conditions (lines 316-317).

⇒ We have modified the text accordingly, on lines 336 and 349-352, removing the comments about drier conditions, and also changed the color bars in Figure 3 to highlight the archeological known periods as in figure 2 (line 824).

Line 2. 'favouring' vs 'favoured' – perhaps past tense best here.

⇒ We have modified the title based on the suggestion of another reviewer. (line 2)

Line 32. The term 'quasi-simultaneously' seems unnecessarily hedging and could be stated more strongly, since the biomarker increases seem to me (eyeballing Figures 2 and S5) that

they are occurring in the same levels. Or include some kind of quantitative estimate of 'within x years' to strengthen this result to be more satisfying than 'quasi-simultaneous'.

⇒ *We have deleted "quasi-" (line 35)*

Line 34. Consider 'wetting' or 'increasing wetness' rather than 'wetter' – see third main comment above.

⇒ *We have modified as suggested for "increasing wetness" (line 37)*

Line 45. Should this be 'the islands of Southeast Asia and Papua'?

⇒ *Yes, we changed it to 'the islands of Southeast Asia and Papua' as suggested (line 50)*

Line 51. I tripped on 'inducing' here – perhaps 'resulting in' or 'motivating'.

⇒ *We replaced inducing with motivating (line 56).*

Lines 54-57. There are more than a few existing climate records, they are just not high quality by 2024 standards. So paucity is not the strongest selling point for this excellent work by the authors. Perhaps re-phrase this section to be more specific about the Pacific climate knowledge gaps filled by this manuscript, like resolution, chronology, multi-proxy, etc.

⇒ *We have rephrased to: "However, the resolution and chronologies of existing climatic records in Remote Oceania covering the period of Lapita settlement do not allow the actual influence of climate on human settlement patterns to be determined." (lines 59-62).*

Lines 58-60. This sentence is dangling in space between paragraphs – does it belong with the next paragraph (lines 61-72)?

⇒ *Indeed, thank you, it belonged to the next paragraph. We have adjusted it.*

Lines 87-89. Any previous sediment work from Emaotfer should be cited somewhere around here, including the initial work by Wirmann whose deeper, older core seems like it was taken very close to the one presented in the manuscript. Excluding other sediment papers from Emaotfer here makes it sound like this is the first time it has been studied, which is not the case.

⇒ *Thank you for pointing this out, we have added references to the work by Wirmann et al. (2011) and Combettes et al. (2015) (line 92-93).*

Lines 117-119. Figure 1C is called out for observable beach ridges at 4m – it's not clear to me where these are in the figure. If this is important, consider improving the figure to make these geomorphological units clear.

⇒ *We have removed the reference to figure 1C as the beach ridges are not evident in this figure. (line 132).*

Line 133. It would be useful to cite the appropriate figure for the C:N (Fig 4) and d15N values (Fig S1?).

⇒ *We have added the citation to both figures (Line 147)*

Line 149. Is the 'drying trend preceding 2900 BP)' truly a drying trend? Compared to later fluctuations it seems somewhere between stable and getting wetter over the previous millennium before 2.9ka (taking the interpretation of depleted C30 d2H shows more rain at face value). Perhaps rephrase.

⇒ *We have rephrased for clarity, using "drier events" (line 163)*

Lines 167-170. This is a good argument for creating an independent chronology across the Lapita period, but totally glosses over the big reversal at 181cm. See second main comment.

⇒ *Please see our reply to your second main comment above.*

Line 173. I tripped over the ordering of dates and periods in this sentence. Perhaps it is better to stay consistent with earliest-to-latest order and change this to '2624-2750 BP to 2739-2879 BP'.

⇒ *We agree with the reviewer's suggestion and have made the changes as suggested (line 200, 208).*

Line 174. Perhaps 'indeed' would be better as 'likely' here given the uncertainties in the ranges reported in this sentence.

⇒ *Modified as suggested (line 201).*

Line 181. Consider change to '2870-2920 BP' for consistency as in Line 173 comment.

⇒ *Modified as suggested (line 208)*

Lines 193-194. Well said.

⇒ *Thank you!*

Lines 197-199. I agree; see comment about Line 32.

⇒ *We have modified the line 35 to be consistent with what expressed here.*

Lines 209-226. This message, especially Lines 213-215, seems rather important. Perhaps this should be highlighted briefly in the abstract.

⇒ *We feel that although important to the thrust of our manuscript, this aspect of our study is better expressed in the main text.*

Lines 261. Perhaps 'manifestations' rather than 'flavours'.

⇒ *We decided to keep the term ENSO flavours as it has been used previously in the literature and is the most commonly used phrasing to differentiate eastern and central Pacific El Nino events e.g. Murphy et al. (2014), Karamperidou and DiNezio (2022), (line 294).*

Line 303. 'increasingly wet' rather than 'wetter'. See third main comment.

⇒ *Modified as suggested (line 336)*

Lines 316-317. This statement is poorly supported by the data presented. The wet period, as interpreted by depleted d2H values, outlasts the palmitone and faecal peak, suggested the end of the Lapita settlement during the transition to the wetter conditions, and gone before the wettest.

⇒ *We have rephrased accordingly (line 349-352).*

Line 319. Regarding the 'more variable and drier climate during the Erueti period', I see the argument for more variable. But would say that the wetness proxy has pretty much the same value for both human periods.

⇒ *We have removed the term "drier" (line 355).*

Lines 363-364. Please detail these plant remains. See second main comment.

⇒ *We have added pictures of the macrofossils that were dated in the supplementary material (Fig. S5).*

Lines 371-375. The details of why these measurements were excluded is useful to evaluate the chronology. But the deeply reversed date right initial human settlement (181 cm, d13C = -28) is glaringly absent from these justifications. See second main comment.

⇒ *The macrofossil at 181 cm was included in the model input (contrary to the macrofossils listed in line 408-410), however the Bayesian age-depth model left this date outside as not coherent with the majority of the younger samples. We have added a sentence to explicitly acknowledge this (lines 416-419).*

Lines 379-381. It would be interesting to see these TIC values somewhere, given the comments on watershed erosion in the manuscript (e.g. Lines 157-158). Perhaps added to Fig 4D, or an expanded Fig. S1.

⇒ *We have added a panel (panel E) to Figure 4 to include TIC% values (line 852).*

Line 716-717. Please cite a source for the Lapita archeological sites mapped.

⇒ *We have added the reference Shaw et al., 2022 (line 787) .*

Lines 719-721. Please cite a source for the geological data mapped.

⇒ *We have added the reference Ash et al., 1978 (line 790).*

Lines 758-759. If the wet period shaded in blue is based on the author's interpretation of the d2H values, then the darker shading should extend to the period of most depleted d2H to around 2.7ka.

⇒ *We have modified the shading to highlight the Lapita and Erueiti period instead (line 831).*

Supplementary Information file

Figure S1. The data plotted do not match the caption. The interpretation of 'marine influence' seems to have the arrow going the wrong direction, since the text argues that higher d15N values reflect oceanic nitrogen sources. It is interesting to note that some of the peat has marine-like d15N. There is a period missing at the end of the caption.

⇒ *Thank you, we have corrected the arrow direction and added the missing period (line S218-227).*

Lines S159-S160. These descriptions of the panels do not match the data presented.

⇒ *Thank you, we have corrected the caption to match the data presented (line S181-182).*

Lines S160-S162. I can see where the authors are going with this, but the colors are just a ramp of depths. This interpretation is stronger based on the crossplot groupings, and it may be more compelling if the cluster of bottom left points were labeled as 'lacustrine' or 'lake' to reflect this interpretation and to be consistent with Figure S2. This sentence could be rephrased to be based on the separations rather than the depths/colors.

⇒ *Thank you, we have modified the caption to reflect the groupings that emerge from the figure (line S221-227).*

Lines S201-S202. Should this title refer to Figure S6 rather than Figure S7?

⇒ *Yes, thank you, this is now corrected as we added the macrofossils pictures in Figure S5.*

Table S1. That is a lot of decimal places for 14C and 13C measurements. Convention is to report 14C years and SD to the nearest year at most. It would be useful to have the postbomb dates also reported as F14C as per convention. One decimal place for d13C is more reflective of accuracy, and what most readers are used to seeing, consider changing. Please provide

taxonomic information for 'plant remains' and 'leaves' and 'seeds', see second main comment. The caption mentions that samples with roots are indicated with an asterisk, but none are provided; please provide.

⇒ *Thank you for your feedback. We have revised Table S1 (line S286) to align with the conventions for reporting ^{14}C and $\delta^{13}\text{C}$ measurements. Specifically, we have:*

1. *Rounded ^{14}C years to the nearest year.*
2. *Included postbomb dates reported as $F^{14}\text{C}$*
3. *Reported $\delta^{13}\text{C}$ values to one decimal place.*
4. *Indicated samples with roots using an asterisk as mentioned in the caption.*

Reviewer #3 (Remarks to the Author):

The manuscript by Camperio et al. provides a novel and interesting study of Lapita agriculture and environmental change via the analysis of fecal sterols and palmitone, which is a biomarker for ingested/digested taro. The samples were obtained from a sediment core and coupled with a robust series of radiocarbon dates, elemental and isotopic analyses of sediments, age-depth modeling, and analyses of alkanolic acids, all of which provide detailed symphony of data that reveal 6000 years of Vanuatu's past. I commend the authors for the design of the research itself, which has much to offer in terms of connecting questions about agriculture and human presence with climate variability. The paper is very well written and sourced, with diligent attention to the presentation of the data in the figures and supplementary tables. I think as a whole the manuscript argues a very strong case for the production and consumption of taro in Vanuatu during the colonization phase of Lapita, and also provides further information about the logistics of that colonization with the related climate and geological data.

The figures are particularly compelling-- I studied them and am satisfied with how they are used to depict the results of the analyses. Overall the structure of the manuscript is well organized, although I found it odd to have the materials and methods follow after the discussion.

⇒ *Thank you for your review. The structure of the journal requests the material and method section to follow the discussion. Therefore we have left them there.*

I think this is a groundbreaking manuscript. I suggest the authors create a title that reflects the analysis of fecal sterols and palmitone, and their use as indicators for taro consumption. Although the results indicating wetter environments at the time of Lapita is important and robustly supported by the analyses, there have been many climate-related Lapita papers over the decades, but none about the molecular data and analyses described here. The manuscript truly stands out--- it deserves a more distinctive and representative title.

⇒ Thank you for the positive feedback! We have modified the title to: “Sedimentary biomarkers of human presence and taro cultivation reveal early horticulture in Remote Oceania”.

Reviewer #4 (Remarks to the Author):

This manuscript presents sedimentological, geochemical, and biomolecular data to assess the timing of human arrival and early horticultural activity on Efate and their relationship with climatic events. I find the paper interesting and an important advancement in the application of biomarkers in archaeological and ecological projects. While I find no issues with the paper’s methods and data presentation, I include here several comments regarding data interpretation and discussion. This is strictly for the authors’ consideration in the hopes of strengthening the paper and I advocate for its publication regardless of the incorporation of these comments.

Comment 1: Highlighting the importance of the Emaotfer fecal stanol record. In my opinion, the stanol data shown here help clarify the difference between nonhuman and human coprostanol signals. The supplemental text gets into this somewhat, but I suggest adding a few sentences to the main text starting on line 171 that acknowledge A) the low level coprostanol values before humans arrive, B) how this demonstrates nonhuman coprostanol production, and C) how this study is uniquely situated to differentiate nonhuman vs. human-related coprostanol production because of the relatively recent and abrupt arrival of humans here that is difficult to find in other areas of the world. I look forward to citing this paper for this very reason and I recommend calling out this aspect of the study in the main text.

⇒ Thank you for your suggestion. We have added a discussion on lines 193-197.

Comment 2: Catchment area: In line 215 the authors write, “The swamp sediment is likely incorporating material from a larger catchment area than what is preserved at the

archaeological site.” Have the authors tried modelling the current catchment area with GIS software? Incorporating a watershed boundary line into Figure 1 might improve the paper.

⇒ We did not try to model the catchment area in this study as forthcoming work on recent lidar imagery has yet to be published and made available by our colleagues (maybe later in 2024). From initial analyses, the currently available Google Earth DEM is very different from these new lidar datasets.

Comment 3: Impact of pigs: In line 217, the authors write, “The biomarkers related to human presence and activity remain high from 2000 BP on, with some variations, indicating continuous human occupation of the island.” How much do nonhuman domesticates, like pigs, have to do with the post-2000 BP disconnect between high coprostanol values and low local archaeological data? Did pigs become feral on Efate? While pigs are discussed in the supplemental text, I believe having at least a sentence that acknowledges potential coprostanol input from nonhuman domesticates is warranted in the main text.

⇒ Thank you for your suggestion, we have modified the sentence on lines 244-247 to acknowledge the potential contribution of pigs to the coprostanol levels.

There is a close relationship between domestic pig production and taro throughout the Pacific. Tethered pigs often occupy fallow cultivation systems. It is highly likely that cultivations ran right up to the edge of the Emoatfer swamp.

However, since palmitone increases simultaneously with coprostanol, and does so even more strongly, the most likely explanation for the observed increase in fecal biomarker is human presence and associated horticulture, as feral pigs would not drive an increase in taro plantation.